# Genetic variations of *CYP2R1* (rs10741657) in Bangladeshi adults with low serum 25(OH)D level—A pilot study

**Kazi Lutfar Rahman**[1]*, **Qazi Shamima Akhter**[2], **Md. Sayedur Rahman**[3], **Ridwana Rahman**[2], **Samina Rahman (Sami)**[4], **Farzana Yeasmin Mukta**[5], **Sudipta Sarker**[6]

**1** Covid Dedicated RT-PCR Lab, Kurmitola General Hospital, Dhaka, Bangladesh, **2** Department of Physiology, Dhaka Medical College and Hospital, Dhaka, Bangladesh, **3** Department of Pharmacology, Bangabandhu Sheikh Mujib Medical University, Dhaka, Bangladesh, **4** Department of Orthopedics, United Hospital Limited, Dhaka, Bangladesh, **5** Department of Cardiology, Kurmitola General Hospital, Dhaka, Bangladesh, **6** Charbaria Union Health and Family Welfare Centre, Barishal Sadar, Barishal, Bangladesh

* klrahman@yahoo.com

**Data Availability Statement:** The relevant data are within the manuscript and its Supporting information files. The detail of the participants is not shared to maintain their privacy concern. All

## Abstract

### Background

Some studies revealed that despite having sufficient sun exposure and dietary supply, the level of serum 25(OH)D in Bangladeshi adults is lower than its normal range. Genetic pattern of an individual is also an essential factor that regulates the level of serum 25(OH)D. However, the genetic variations of *CYP2R1* (rs10741657) and their association with low serum 25(OH)D level in Bangladeshi adults are yet to be explored.

### Objective

This study was conducted to determine the frequency of variants of rs10741657 of *CYP2R1* gene and its association with low serum 25(OH)D level among Bangladeshi adults.

### Method

This pilot study was conducted among thirty individuals with low serum 25(OH)D level as the study population and ten subjects with sufficient serum 25(OH)D level as controls based on the inclusion and exclusion criteria. Genetic analysis of rs10741657 of *CYP2R1* including primer designing, DNA extraction, PCR of target region with purification and Sanger sequencing of the PCR products were done accordingly. For statistical analysis, One-way ANOVA followed by LSD test, Freeman-Halton extension of Fisher's exact test, Chi-square test ($\chi^2$) test and unpaired student t-test were performed.

### Results

In this study, genetic variants of *CYP2R1* (rs10741657) among the study population were genotype GG (63.30%), GA (30%) and AA (6.7%). Minor allele frequency of the study population was 0.217. The association between GG and GA genotypes of CYP2R1 (rs10741657) with low serum 25(OH)D level among the study population was found and it

Patients data are now uploaded in a file named "Available Data of the study subjects".

**Funding:** This study was partially funded by Bangladesh Medical Research Council (BMRC). The funder had no role in study design, data collection and analysis, decision to publish, or preparation of the manuscript.

**Competing interests:** The authors have declared that no competing interests exist.

was statistically significant. Statistically significant differences were also observed between the genotypes and alleles of the study population and controls.

## Conclusions

The presence of 'GG' and 'GA' genotypes of rs1041657 in *CYP2R1* gene is associated with low serum 25(OH)D level among Bangladeshi adults in this pilot study.

## Introduction

In human physiology vitamin D is appreciated as a very important nutrient since its identification. The structure of vitamin $D_3$ was first determined in 1936 and its activation process was established in 1971 [1]. Over the last few decades, it has been proven that vitamin D is an essential nutrient for a number of physiological functions [1, 2]. Inadequacy of vitamin D level in human body is associated with rickets and osteomalacia as well as a wide range of other illnesses like various cancers and several autoimmune diseases, psychological and cardiovascular disorders. It has been observed that about 1 billion people have vitamin D deficiency or insufficiency all over the world [3]. Vitamin D deficiency is a condition when the serum 25(OH)D level is below 20 ng/ml and vitamin D insufficiency is when the serum 25(OH)D level is 21–29 ng/ml [4]. Both these conditions are termed as low serum 25(OH)D level [5]. According to various studies around 51–100% of the different groups of population suffered from low vitamin D level [3, 6–8]. In spite of adequate sun exposure, a recent study showed that about 70.7% people living in coastal area of Cox's Bazar, Bangladesh had low serum 25(OH)D level [9].

Sunlight (UVB exposure), seasonal variation, skin color, age, race, gender, BMI and dietary/supplemental vitamin D intake are some of the major factors that influence serum vitamin D level of an individual [10]. Genetic pattern is also one of the key factors that may cause variable serum vitamin D level [11]. In recent times, a number of molecular conceptions regarding vitamin D level have been added to genome-wide association studies [12]. It has been revealed that genetic factors may affect the bioavailability of vitamin D in the circulation of human being ranging from 53% to 68.9% [13]. The presence or absence of various single nucleotide polymorphisms (SNPs) in vitamin D related genes is one such factor. The common vitamin D related genes are–*CYP2R1, DHCR7, DBP, VDR, CYP24A1 and CYP27B1*. Among these, *CYP2R1* (cytochrome P450 family 2 subfamily R polypeptide 1) encodes for 25-hydroxylase enzyme. This enzyme is responsible for the formation of 25(OH)D in liver [13, 14].

It has been observed that variants of rs10741657 in *CYP2R1* gene may give rise to low serum 25(OH)D level irrespective of adequate sun exposure and healthy diet [15]. Despite abundant sun exposure low serum 25(OH)D level has been found among most Bangladeshi people [8, 9]. The genetic influence of rs10741657 of *CYP2R1* on low serum 25(OH)D level has never been investigated before in Bangladesh.

Hence, the goal of the study was to assess the frequency of the genetic variants of *CYP2R1* (rs10741657) in Bangladeshi adults with low serum 25(OH)D level and also in Bangladeshi adults with sufficient serum 25(OH)D level. Additionally, possible associations between the genotypes of *CYP2R1* (rs10741657) and serum 25(OH)D level of the adults with low serum 25(OH)D level and also with those of controls were investigated.

## Materials and methods

### Study design

This cross-sectional study was carried out after obtaining the ethical clearance from the Research Review Committee of the Department of Physiology and Ethical Review Committee of Dhaka Medical College, Dhaka, Bangladesh from July 2019 to June 2020. Permission was taken from respective authorities of the Center for Medical Biotechnology, MIS (Management Information System), DGHS (Directorate General of Health Service), Dhaka, Bangladesh, for the genetic analysis and Dhaka Metropolitan Police, clubs of Abahani field, Dhaka and Good neighbors Bangladesh, an NGO, for collecting the samples (S1 Appendix). To conduct this study 18–60 years old Bangladeshi middle class (assessed by Kuppuswamy socio-economic scale [16]) males and females (non- veiled) with light brown complexion (assessed by Fitzpatrick scale), normal BMI (18.5–24.9 kg/m$^2$) and sun exposure of at least 45 minutes between 11 am to 2 pm were selected. Umbrella or sun block users, smokers, pregnant or lactating mothers and people with pathological conditions like chronic liver disease, chronic renal disease, malabsorption syndrome, chronic pulmonary disease; systemic disease like Diabetes mellitus were excluded from this study. People taking vitamin D or calcium supplements, estrogen or progesterone replacement therapy or medications such as steroid (5mg/day for last 3 months), anticonvulsant, thiazide diuretics, antifungal agents were also excluded. Those exclusions were done based on their history of chronic diseases and estimation of their serum creatinine, glutamic pyruvic transaminase level, prothrombin time and random blood glucose. Serum calcium and albumin level were also estimated to exclude any pathological condition related to serum calcium and albumin level. All the study subjects were healthy individuals with no other medical problems. Finally, Traffic Police, Community NGO workers (Health Workers), Cricket and Football players were included as the study subjects. Prior to the collection of their blood samples, all participants were informed about the study process and the written consent were taken.

### Study subjects

Since it was a pilot study the subject numbers were kept at 30 adults with low serum 25(OH)D level as study population and 10 controls with sufficient serum 25(OH)D level for the analysis of rs10741657 of *CYP2R1* gene. The level of Serum 25(OH)D <20 ng/ml, 21–29.9 ng/ml and ≥30 ng/ml were considered as serum 25(OH)D level deficiency, insufficiency and sufficiency accordingly and both insufficiency and deficiency were considered as low serum 25(OH)D level for the current study. Blood sample collection was done until the accomplishment of the sample size. As a result, a total of 54 blood samples were collected in the beginning of the study. Based upon the investigation reports of serum 25(OH)D and other parameters, 2 subjects were excluded for having higher serum creatinine level than normal. Among the rest, 30 subjects were found with low serum 25(OH)D level (<30 ng/ml) whereas the other 22 were found with sufficient serum 25(OH)D level (≥30 ng/ml). Finally, 30 subjects with low serum 25(OH)D level were selected as the study population and 10 subjects with sufficient serum 25 (OH)D level were selected as the controls for comparison. The controls were selected from those 22 samples with sufficient serum 25(OH)D level using a random table of Microsoft excel 2010 software.

### Biochemical analysis

Serum 25(OH)D levels of the participants were estimated by chemiluminescence immunoassay using MAGLUMI series Fully- auto chemiluminescence immunoassay analyzer (intra-assay CV 4.99%, inter-assay CV 5.93%). Serum calcium, creatinine, albumin and SGPT were

measured in Automated Biochemistry Analyzer. Measurement of random blood glucose was done by Hexokinase method in Automated Biochemistry Analyzer and Prothrombin time (PT) was analyzed on automated coagulation analyzer. All those biochemical analyses were performed at the department of pathology, Dhaka medical college, Dhaka, Bangladesh.

## Molecular analysis

For genetic analysis of rs10741657 of *CYP2R1* the blood samples were carried to Center for Medical Biotechnology, MIS, DGHS, Dhaka, Bangladesh and stored at -70ºC freezer. Genomic DNA was extracted and isolated from the samples using Promega reliaPrepTM Blood gDNA isolation kit (Promega, USA). After taken out from the freezer, the frozen blood samples thawed completely at room temperature. Proteinase K enzyme was used to break the cellular proteins of the samples. Cell lysis buffer was added after that to breakdown the cells and the DNA came out in this process. Then the samples were kept at 56ºC in a heat block machine for 10 minutes. Binding buffer was mixed to bind with the DNA of the samples and column wash solution was applied to wash away the waste thereafter. Finally, the DNA of the samples were stored with nucleus free water at -20ºC freezer. The information of selected SNP rs10741657 was taken from NCBI database. The website, "https://wheat.pw.usda.gov/demos/BatchPrimer3/" was used for primer designing. Then it was validated by 'IDT OligoAnalyzer Tool' of Integrated DNA Technologists. The polymerase chain reaction (PCR) of rs10741657 was performed using forward and reverse primers:

`5'-TTCAATAATCAGAAGCAAACAAAAAGTGC-3'`

and `5'-GGTTTTAAGCCATCAGATTGGTGGTAAT-3'`.

Polymerase Chain Reaction (PCR) of extracted DNA was done using GoTag G2 Green PCR master mix (Thermofisher) and the designed primers of rs10741657. The whole procedure was performed under a biological safety cabinet to prevent DNA contamination. Several attempts were taken to achieve a good PCR product. The volume, temperatures, PCR cycles and DNA amount for a good PCR product were optimized. Finally, Polymerase Chain Reaction of a volume of 30 μl solution was done consisted of 15 μl PCR master mix, 1.5μl(50ng) extracted genomic DNA, 075 μl forward primer, 0.75 μl reverse primer and 12 μl nucleus free water. PCR reaction profile was carried out by initial denaturation at 95˚C for 3 min, followed by 32 cycles of denaturation at 95˚C for 30 sec, annealing at 58.04˚C for 30 sec and a final extension at 72˚C for 5 min. After that, Gel electrophoresis was done to check the PCR product whether the desired PCR product was present or not. Then all the PCR products were purified by QIAquick PCR Product Purification Kit (QIAGEN PCR Product Purification Kit). The Sequencing of the purified PCR products was performed by ABI Sanger Sequencer 3500 device. The ABI-prism Bigdye terminator version 3.1 sequencing ready reaction kit was used in cycle sequencing. For data analysis, the AB1 files generated by Sanger Sequencer were imported to the Chromas software using Chromas® Software version 2.66. The chromatogram data was then edited and aligned to find out the good quality data and saved as SCF file. For interpretation of the mutation data, query sequence was compared with available known BLAST database and the SNP has been identified. Then the saved SCF files were converted to FASTA format for the use in Mega 7 software. All the final sequencing data of the study subjects were uploaded and published on NCBI GenBank (S2 Appendix).

## Statistical analysis

Allele and Gene frequencies with a percentage of rs10741657 were analyzed among the study population (N = 30) and the controls ($N_c$ = 10). Statistical analysis to see the association

between low serum 25(OH)D level and the genotypes of *CYP2R1* (rs10741657) in the study population was performed by One way ANOVA followed by LSD test. Freeman-Halton extension of Fisher Exact Test was done to compare between the genotypes, whereas Chi-squared Test was done to compare between the alleles of the study population and controls. Unpaired student t-test was performed to compare between sociodemographic and biochemical variables of the study population and controls. A difference was considered as statistically significant if *p* value was <0.05. Statistical analysis was done by using a computer based statistical program SPSS version 25.0.

## Result

The socio-demographic characteristics and biochemical parameters of the study population and controls are shown in Tables 1 and 2, respectively. Among the study population and controls, all other parameters were within the normal ranges except serum 25(OH)D level (Table 2).

### Genotypes of rs107416567 of *CYP2R1* gene

The genotyping data of rs10741657 of *CYP2R1* of the study subjects were derived from Sanger sequencing (Figs 1–3). There were 3 variants of genotype in rs10741657 of CYP2R1 gene: genotype GG, GA and AA.

Among the study population the mean difference of serum 25(OH)D level of the variant genotype groups (GG, GA and AA) was statistically significant. The mean serum 25(OH)D level of genotype group GG was lower than group GA and group AA. The serum 25(OH)D level was also found to be lower in genotype group GA than group AA. The mean difference of serum 25(OH)D level among the genotype group GG and GA was statistically significant. So, it can be emphasized that the genotype GG and GA of rs10741657 were associated with low serum 25(OH)D level among the study population and there was no association of any of the genotypes of rs10741657 with sufficient serum 25(OH)D level among the controls (Table 3).

**Table 1. Socio-demographic characteristics of the study subjects ($N_t$ = 40).**

| Parameters | | Study subjects (N = 30) | Controls ($N_c$ = 10) | *P* value |
|---|---|---|---|---|
| Age (Years) | | 34.60±12.87 | 28.90±9.26 | 0.205 [ns] |
| Sex | | | | |
| | Male | 25 (83.30%) | 07 (70%) | 0.374 [ns] |
| | Female | 05 (16.70%) | 03 (30%) | |
| BMI (kg/m$^2$) | | 22.59±1.13 | 22.23±1.02 | 0.374 [ns] |
| SBP (mmHg) | | 116.33 ± 7.18 | 114.50 ± 4.37 | 0.45 [ns] |
| DBP (mmHg) | | 75.50 ± 6.34 | 77.00 ± 5.86 | 0.5136 [ns] |
| Occupation | | | | |
| Traffic Police | | 14(46.70%) | 02(20%) | 0.066 [ns] |
| Player | | 11(36.70%) | 05(50%) | |
| NGO worker | | 05(16.70%) | 03(30%) | |

Results were expressed as mean ±SD, frequency and percentage; % = Percent; BMI = body mass index; SBP = systolic blood pressure; DBP = diastolic blood pressure; N = total number of study population; $N_c$ = total number of controls; Figure in parenthesis shows percentage. Blood pressure was measured manually by stethoscope and sphygmomanometer during data collection; Unpaired student t-test was performed to compare between two groups. The test of significance was calculated for all comparisons and *p* value <0.05 was accepted as level of significance.

[ns] = not significant,

* = significant.

**Table 2. Biochemical parameters of the study subjects ($N_t$ = 40).**

| Parameters | Study subjects (N = 30) | Controls ($N_c$ = 10) | *P* value |
|---|---|---|---|
| Serum 25(OH)D (ng/ml) | **25.22±2.15** | **49.64±15.25** | **0.0001*** |
| Serum Albumin (gm/dl) | 4.34±0.40 | 4.06±0.30 | 0.05 ns |
| Fasting blood glucose (mmol/L) | 4.49±0.51 | 4.84±0.36 | 0.05 ns |
| SGPT (U/L) | 17.53±5.56 | 22.60±6.09 | 0.02* |
| Serum Creatinine (mg/dl) | 1.048±0.18 | 0.99±0.174 | 0.461 ns |
| Prothrombin time (second) | 12.60±0.85 | 13.20±1.54 | 0.017* |
| Serum Calcium (mg/dl) | 9.10±0.39 | 9.51±0.40 | 0.007* |

Results were expressed as mean and standard deviation (mean ± SD); 25(OH)D = 25-hydroxy vitamin D;
SGPT = serum glutamic pyruvic transaminase; N = total number of study population; $N_c$ = total number of controls.
Unpaired student t-test was performed to compare between two groups. The test of significance was calculated for all
comparisons and *p* value <0.05 was accepted as level of significance.
ns = not significant,
* = significant.

There was a statistically significant difference between the variant genotypes of rs10741657 of
*CYP2R1* gene of the study population and the controls (p = 0.033) (Table 4). In addition, it was
also found that in the study population the distribution of allele G was 78.3% whereas of allele
A was 21.7%. While in the control group, the distribution of allele G was 45% and that of allele
A was 55%. This showed a statistically significant difference between the alleles of rs10741657
of *CYP2R1* gene in the study population and the controls (p = 0.010).

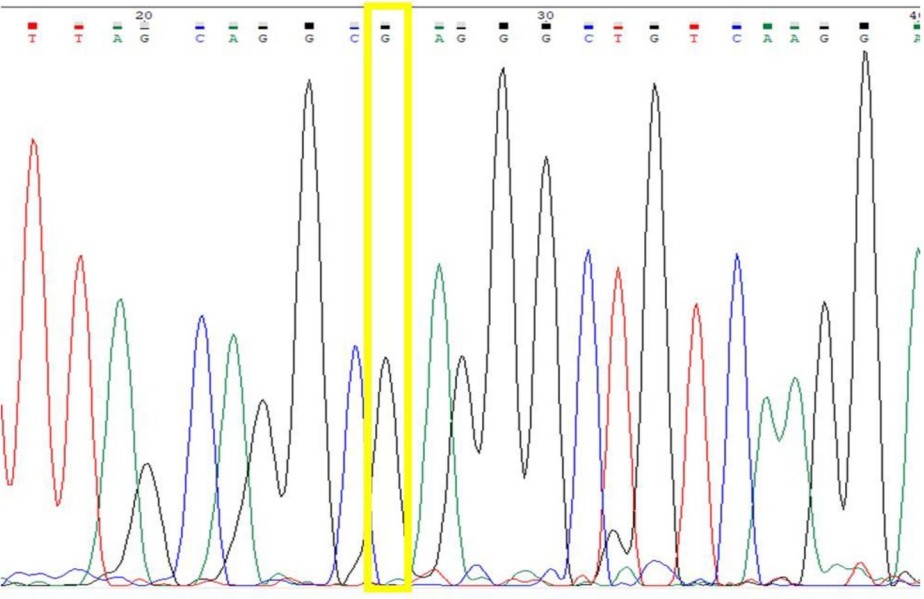

Raw data derived from Sanger sequencer. Samples were edited by Chromas® Software version

2.66

**Fig 1. Sample of chromatogram data of rs10741657 showing G allele in location 26 bp.**

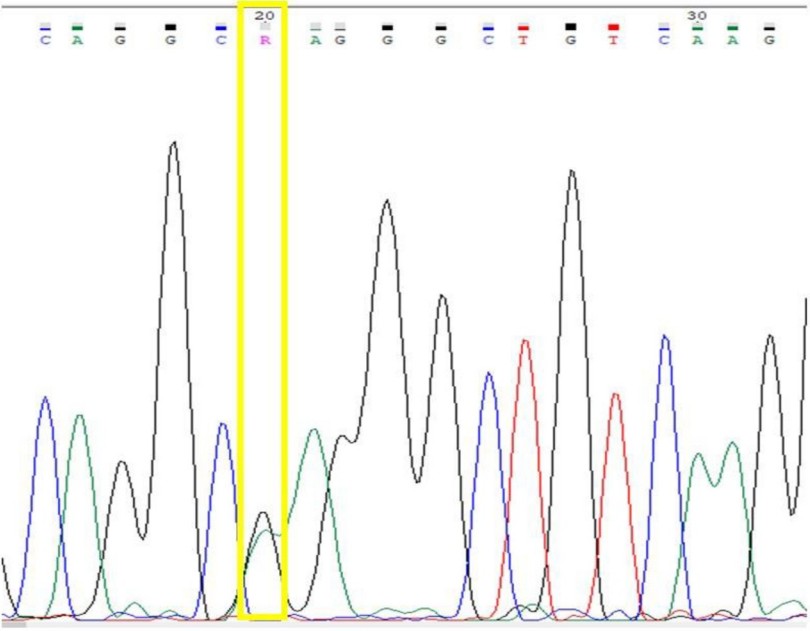

Raw data derived from Sanger sequencer. Samples were edited by Chromas® Software version

2.66

**Fig 2. Sample of chromatogram data of rs10741657 showing the SNP- R (heterozygous G+A) allele in location 20 bp.**

## Discussion

The present pilot study revealed that GG, GA, and AA are the variants of *CYP2R1* (rs10741657) in Bangladeshi adults. Similar variants of rs10741657 in *CYP2R1* gene were found in studies from different countries like Saudi Arabia, Singapore, Jordan, and other countries [10, 13, 15, 17, 18]. The rs10741657 of *CYP2R1* gene was found to be a predictor of serum 25(OH)D level in several other studies [19, 20]. This rs10741657 was associated with low level of serum 25(OH)D in Caucasian subjects [21]. Peterson et al. [22], had performed a longitudinal study in Denmark where they found the children carrying genotype GG and GA of rs10741657 in *CYP2R1* were associated with low serum 25(OH)D level during the Winter and Spring, whereas it was interestingly observed that children with genotype AA always had the higher mean serum 25(OH)D level in all three seasonal cycles. In addition, Zhang et al.'s [11] study also identified GG genotype in rs10741657 to be associated with low serum 25(OH) D level. These findings were consistent with our study as well, where we found that the maximum number of the people in our study population who had low serum 25(OH)D level were with the genotypes GG and GA. On the contrary, in a study which was conducted amongst Korean population, it was found that there was no association of serum 25(OH)D level with any variants of rs10741657 in *CYP2R1* gene [23]. Thereby this lends us to think that there could be some role of natural selection in the process amongst different ethnicity.

rs10741657 is an important site located at the promoter region of *CYP2R1* gene which encodes 25-hydroxylase enzyme. Variation of rs10741657 can affect the rate and activity of this enzyme. Since it was found in the current study that the percentage of study population

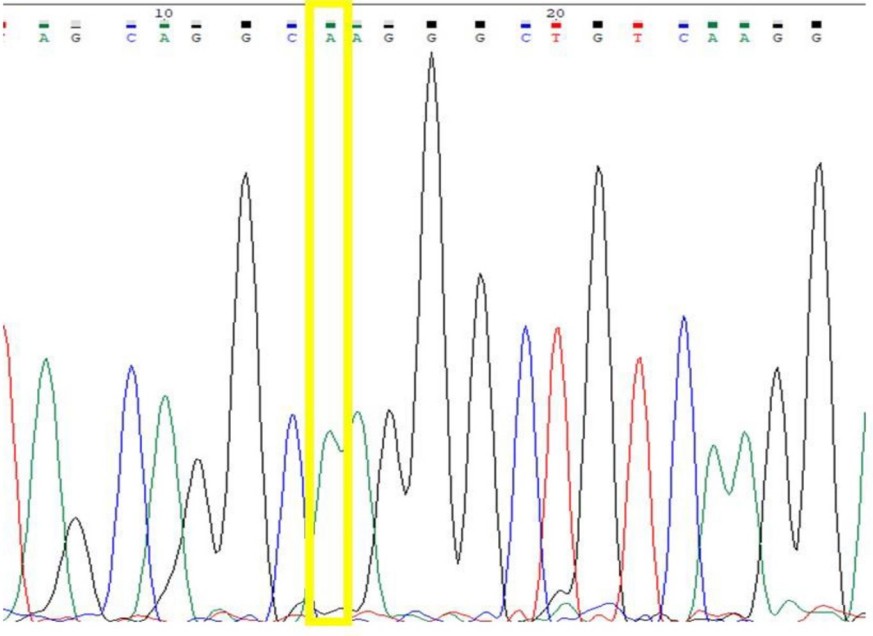

Raw data derived from Sanger sequencer. Samples were edited by Chromas® Software version

2.66

**Fig 3. Sample of chromatogram data of rs10741657 showing the SNP- A allele in location 14 bp.**

having genotypes GG and GA to be higher than those with genotype AA, so there might be a possibility that the presence of "G allele" instead of "A allele" in rs10741657 plays a role in reducing the activity or rate of the production of *25-hydroxylase* enzyme. In one study a group of healthy Caucasian subjects with allele G were found to have low serum 25(OH)D level despite six months of artificial UVB therapy with a slight increase from their baseline [24]. Similarly, another study also found that allele G (GG and GA variants) of rs10741657 had minimum response whereas serum 25(OH)D level increased by 2.5 times in genotype AA after 9 weeks of vitamin D supplementation among Iranian adolescent girls [25]. So, it can be hypothesized that the presence of G allele (Genotype GG and GA) of rs10741657 may have a role in the presence of low level of serum 25(OH)D among individuals. However, the mechanistic aspect of allele G's influence on 25-hydroxylase enzyme and subsequent serum level of 25(OH)D is still not understood clearly.

Based on the ethnicity of this study, the minor allele of rs10741657 in *CYP2R1* gene of the study population was A and the minor allele frequency was 0.217. This was different in Jordanian (0.35), Caucasian (0.42), Lebanese (0.29), Virginian (0.42), Arabian (0.33) and South Asian (0.36) people as per different other studies [15, 17, 18]. According to the Genbank data of NCBI, the higher percentages of A allele (minor allele) were found among the group of people living in northern latitudes like- Northern Sweden (0.48), Estonia (0.42), Korea (0.40), UK (0.39) compared to the tropical and subtropical countries [26]. So, there might be a possibility of natural selection where people from the tropical and sub-tropical region carry higher

**Table 3. Serum 25(OH)D level among the genotypes of rs10741657 in *CYP2R1* gene of the study population (N = 30) and Controls (N$_c$ = 10).**

| Serum 25(OH)D level among the genotypes of rs10741657 among the study population | | |
|---|---|---|
| **Genotype (n) vs Genotype (n)** | **Serum 25(OH)D level (ng/ml)** | ***p*-value** |
| **GG (19) vs GA (09) vs AA (02)** | 24.46±1.18 vs 26.51±2.22 vs 26.63±1.80 | 0.033* |
| **GG (19) vs GA (09)** | 24.46±1.18 vs 26.51±2.22 | 0.016* |
| **GA (09) vs AA (02)** | 26.51±2.22 vs 26.63±1.80 | 0.937 ns |
| **GG (19) vs AA (02)** | 24.46±1.18 vs 26.63±1.80 | 0.149 ns |
| Serum 25(OH)D level among the genotypes of rs10741657 among the controls | | |
| **Genotype (n) vs Genotype (n)** | Serum 25(OH)D level (ng/ml) | *p*-value |
| **GG (02) vs GA (05) vs AA (03)** | 37.49±2.48vs 48.60±15.82 vs 59.46±15.68 | 0.314 ns |
| **GG (02) vs GA (05)** | 37.49±2.48 vs 48.60±15.82 | 0.395 ns |
| **GA (05) vs AA (03)** | 48.60±15.82 vs 59.46±15.68 | 0.344 ns |
| **GG (19) vs AA (02)** | 37.49±2.48 vs 59.46±15.68 | 0.145 ns |

Results were expressed as Mean ±SD. One way ANOVA followed by LSD test was performed to compare between groups; N = Total number of study population, n$_{GG}$ = number of genotype GG, n$_{GA}$ = number of genotype GA, n$_{AA}$ = number of genotype AA; The test of significance was calculated for all comparisons and *p* value <0.05 was accepted as level of significance.

ns = not significant,

* = significant.

**Table 4. Comparison of serum 25(OH)D level and genotypes of the study subjects (N$_t$ = 40).**

| Serum 25(OH)D Level | Genotypes of rs10741657 in CYP2R1 gene | | | *p-v*alue |
|---|---|---|---|---|
| | **GG (n$_{tGG}$ = 21)** | **GA (n$_{tGA}$ = 14)** | **AA (n$_{tAA}$ = 5)** | |
| Subjects with low serum 25(OH)D (N = 30) | 19 (63.3%) | 09 (30%) | 02 (6.7%) | 0.033* |
| Controls with sufficient serum 25(OH)D (N$_c$ = 10) | 02 (20%) | 05 (50%) | 03 (30%) | |

Freeman-Halton extension of Fisher's exact test was done to analyze data; N$_t$ = Total number of subjects; N = Total number of study population; N$_c$ = Total number of controls; n$_{tGG}$ = Total number of genotype GG in study subjects; n$_{tGA}$ = Total number of genotype GA in study subjects; n$_{tAA}$ = Total number of genotype AA in study subjects; Figure in parenthesis shows percentage; The test of significance was calculated for all comparisons and *p* value <0.05 was accepted as level of significance,

* = significant.

percentages of allele G and lower percentages of allele A compared to the people from the countries with shorter duration of daylight.

Our results could potentially be explained further by the aspect of the geographical location of Bangladesh. Bangladesh is located at the latitude of 20˚34' to 26˚38' North and 88˚01' to 92˚41' East longitude. The sun exposure is sufficient and people living here are getting adequate sunlight [27]. In this present study the study population and control group both received adequate sun exposure on daily basis and on their socio-economic status was homogenous and other biochemical parameters were within normal range. It was also observed that the frequency of GG was more frequent than the frequency AA among the study population, which could be an instance where this variation of the genetic makeup of the CYP2R1 (rs10741657) among Bangladeshi adults is influenced by the process of natural selection. The study population had values within normal range for serum calcium level along with some other biochemical markers (SGPT, PT, FBS, Serum Creatinine) and those parameters were also normal in the

control group. As the study population had low serum 25(OH)D level despite the above-mentioned biochemical markers being in normal range and they did not exhibit any history of low Vitamin D related disorders, it can be assumed that the low serum 25(OH)D level of study population could naturally be adequate to perform their physiological functions.

There could be a recommendation to carry out similar study with a larger sample size while observing other biochemical markers (like Parathyroid hormone), as well across other tropical and sub-tropical countries including Bangladesh to assess the influence of other vitamin D related gene variations to reach a more definitive conclusion. Thereby it would be advisable to do a review of the reference values of serum 25(OH)D level used in medical science for the assessment of serum vitamin D level and tailor it to different ethnicity and geographies accordingly. Correspondingly, it could be advisable to conduct the routine investigation for polymorphisms of Vitamin D related genes among individuals. The implication of this study in medical science could be that the people with polymorphic vitamin D related genes may have different reference values for their physiological assessments.

## Limitation

Genetic analysis of other reference sequencing numbers of various genes which might be associated with low serum 25(OH)D level of CYP2R1 gene could not be done due to time and financial constraints. Moreover, it was a pilot study in Bangladesh and we could not include a large number of study population.

## Conclusion

This pilot study has demonstrated that the presence of 'GG' and 'GA' genotypes of rs10741657 of *CYP2R1* are associated with low serum 25(OH)D level among Bangladeshi adults. From this study we can anticipate that the requirement of serum 25(OH)D level could vary from individual to individual to perform their physiologic functions. Since the study sample size was small and other vitamin D related gene variations could not be considered, it remains to be elucidated further through a more robust and expansive study including different regions of Bangladesh to definitively conclude and work towards considering to redefine reference values of serum 25(OH)D level for Bangladeshi adults in the field of medical science for more tailored approach for the diagnosis and treatment purpose in Bangladesh.

## Supporting information

**S1 Appendix. Ethical committee and corresponding permissions.**
(DOCX)

**S2 Appendix. NCBI extension numbers of sequence files.**
(DOCX)

**S1 File. Available data of the study subjects.**
(DOCX)

## Acknowledgments

We would like to convey our heartfelt appreciation for the participants' assistance. We are also thankful to Marufur Rahman Opu, Deputy Program Manager and Md. Ruhul Amin, Medical Office from Center for Medical Biotechnology, Management Information System, Directorate General of Health Services (DGHS), Dhaka, Bangladesh for their supports and encouragements throughout the genetic laboratory works. Mohammad Morshad Alam, Consultant,

Ministry of Health and Family Welfare, World Bank Bangladesh and Khan Tanjid Osman, Postdoctoral Associate, Massachusetts Institute of Technology, Cambridge, Massachusetts, US, for their remarkable guidance and helpful suggestions behind this research work and Dhaka Metropolitan Police, Dhaka, Bangladesh for granting the permission to include the traffic polices of Dhaka city as the study subjects in this research.

## Author Contributions

**Conceptualization:** Kazi Lutfar Rahman, Qazi Shamima Akhter.

**Data curation:** Kazi Lutfar Rahman.

**Formal analysis:** Kazi Lutfar Rahman, Md. Sayedur Rahman, Samina Rahman (Sami), Farzana Yeasmin Mukta, Sudipta Sarker.

**Funding acquisition:** Kazi Lutfar Rahman.

**Investigation:** Kazi Lutfar Rahman.

**Methodology:** Kazi Lutfar Rahman, Qazi Shamima Akhter.

**Project administration:** Kazi Lutfar Rahman.

**Resources:** Kazi Lutfar Rahman.

**Software:** Kazi Lutfar Rahman, Sudipta Sarker.

**Supervision:** Kazi Lutfar Rahman, Qazi Shamima Akhter.

**Validation:** Kazi Lutfar Rahman, Qazi Shamima Akhter.

**Visualization:** Kazi Lutfar Rahman, Qazi Shamima Akhter.

**Writing – original draft:** Kazi Lutfar Rahman.

**Writing – review & editing:** Kazi Lutfar Rahman, Qazi Shamima Akhter, Md. Sayedur Rahman, Ridwana Rahman, Sudipta Sarker.

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
