## [Decision Letter · Decision Letter 0]

20 Sep 2021

PONE-D-21-25624Genetic Variations of CYP2R1 (rs10741657) in Bangladeshi adults with low serum 25(OH)D levelPLOS ONE

Dear Dr. Rahman,

Thank you for submitting your manuscript to PLOS ONE. After careful consideration, we feel that it has merit but does not fully meet PLOS ONE’s publication criteria as it currently stands. Therefore, we invite you to submit a revised version of the manuscript that addresses the points raised during the review process. Besides the important reviewers' comments, authors should address the following points:Please, follow appropriate reporting guidelines, such as STROBEPlease, better specify the objectives of the study; it would be better to add the word to determine “frequency” of variants…)The small sample size is a major limitation. Authors stated that this is a pilot study (this should be better also included in the title, discussion, and conclusion).Regarding Data availability, authors stated “Yes - all data are fully available without restriction” and data are “within the manuscript and its as supporting information files”. However, the submission doesn’t contain unidentified patients’ data. According to PLOS ONE publication criteria, authors are required to make all data underlying the findings described fully available, without restriction, and from the time of publication. PLOS allows rare exceptions to address legal and ethical concerns, that should be clearly explained.Authors should clearly describe participants’ inclusion and exclusion criteria, time of recruitment, how participants were selected (convenience sample?)Authors should provide definitions/diagnostic criteria for described conditions/outcomes, including vitamin D deficiency. The authors didn’t study other genetic determinants (other vitamin D associated genes), which could be possible confounders.In statistical analysis, were data tested for normality (some data are reported as mean (SD) but appear not normally distributed). What about confounders?Study results and tables should be reported in the results section.Tables require better organization with comparison of variables between the study and control groups.In figure 1, it would be better to annotate the polymorphisms.Figures 2 & 3 seem to be of little value as its data can be incorporated in tables or text.Discussion requires major revision, critically discussing main findings of study (without repeating results) and study implications.In light of being a pilot, the current conclusion seems to overestimate study findings; authors should reduce the strength of their statement, confirming that this is a pilot study, and recommend further well-powered studies.Please, revise the references to follow PLOS ONE style.Authors should revise the manuscript for several errors in language, sentence structure, grammar, and punctuation.

We look forward to receiving your revised manuscript.

Kind regards,

Elsayed Abdelkreem, MD, PhD

Academic Editor

PLOS ONE

Journal Requirements:

2. Please ensure that you refer to Figure 3 in your text as, if accepted, production will need this reference to link the reader to the figure.

Reviewers' comments:

Reviewer's Responses to Questions

**Comments to the Author**

1. Is the manuscript technically sound, and do the data support the conclusions?

Reviewer #1: Partly

Reviewer #2: Yes

Reviewer #3: Yes

2. Has the statistical analysis been performed appropriately and rigorously? 

Reviewer #1: I Don't Know

Reviewer #2: Yes

Reviewer #3: I Don't Know

3. Have the authors made all data underlying the findings in their manuscript fully available?

Reviewer #1: Yes

Reviewer #2: Yes

Reviewer #3: Yes

4. Is the manuscript presented in an intelligible fashion and written in standard English?

Reviewer #1: Yes

Reviewer #2: Yes

Reviewer #3: No

5. Review Comments to the Author

Reviewer #1: I have a major concern regarding the sample size of the study. With the current small sample size and the used study design, the study probably is underpowered. Was the sample size and power of the study was calculated prior starting the study?

There are other points to consider as follows:

1. In line 73 (in the introduction), Please remove DBP as GC is already mentioned and It is the gene encoding vitamin D binding protein (DBP). In addition, please include CYP27B1 gene with the other major common vitamin D genes as it is one of the significant genes involved in vitamin D metabolism (it is the gene encoding 1α-hydroxylase, the enzyme responsible for the 2nd step of activation of vitamin D in the kidney).

2. Please mention the exclusion and inclusion criteria in the methods and the inter and intra-assay CV for the performed tests.

3. Please move Table 1 and 2 to the Results section.

4. Information in line 186-187 (in the results) is already mentioned in the methods, please remove.

Reviewer #2: First of all, I would like to congratulate the researchers for this very good work. The "CONCLUSION" section is also finished with a very good conclusion. However, the deficiency that I see in the study and other similar studies is that it is in the form of statistically researching the negativities associated with vitamin D deficiency and making a connection between them. This should provide an advantage if a selective mutation is dominant in geographic regions where this mutation is present. What advantage does vitamin d provide at lower levels? What protection does it provide for people in those areas? The answer to this question should also be explored in the comments. Natural selection never happens without a reason. When I look at it from the point of view of the country I live in, kidney stones come to mind. But I think this should be investigated. If researchers can add an answer and comment to this question in terms of their own country, I think that their work will be more beneficial than other studies. Beautiful original work worth publishing. The journal meets all publication criteria. I wish the researchers success in their new work.

Reviewer #3: Thank you for your work. Some comments

1. Grammar and sentence construction i nthe entire manuscript have to be revised/ improved

2. What is the significance of yorus tidu if this has been reported in the literature already?

3.inclusion and exclusion criteria not stated

4. how di you come up with the final nuber of subjects and how did you choose the initial of 54. please state clearly how you got yoru subjects

5. what was the significantce of all the other biochemical parameters that were taken apart form vitamin D levels

6. for tables 1 and 2 make them reader friendly-remove too many lines and make it cleaner. yo ulaso did not specify the statistical significance betwenn the two groups of all the parameters

7. for table 3 you did not compare the results between groups. you only compared them within the groyp. it wuld be ebtter if you compared for exampe the genoype GG of affected and controls and the corresponding statistical significance

8. i nthe discussion, do not repeat anymore your results but instead interpret and explain them. your tables hsould be self explanatory and in the discussion you will only give its importance

9. what is the impact of the study?

10. what is the implciation i nterms of treatment?

11. make the discussion mroe concise

6. PLOS authors have the option to publish the peer review history of their article (what does this mean?). If published, this will include your full peer review and any attached files.

Reviewer #1: **Yes: **Shatha Alharazy

Reviewer #2: No

Reviewer #3: No

---

## [Author Response · Author response to Decision Letter 0]

29 Sep 2021

Responses to academic editor

• Please, follow appropriate reporting guidelines, such as STROBE

Response: Thank you for your kind concern. In this 1st revised version of the manuscript, we have tried to maintain it accordingly.

• Please, better specify the objectives of the study; it would be better to add the word to determine “frequency” of variants…)

Response: This has been done in the current version of the manuscript. Please check page no. 2, line no. 33 and page no. 2, line no. 84 to 88.

• The small sample size is a major limitation. Authors stated that this is a pilot study (this should be better also included in the title, discussion, and conclusion).

Response: Thank you so much, it was really needed. This has been done in the current version of the manuscript. Please check page no. 1, line no. 5, and page no. 13, line no. 258 and page no. 16, line no. 326.

Regarding Data availability, authors stated “Yes - all data are fully available without restriction” and data are “within the manuscript and its as supporting information files”. However, the submission doesn’t contain unidentified patients’ data. According to PLOS ONE publication criteria, authors are required to make all data underlying the findings described fully available, without restriction, and from the time of publication. PLOS allows rare exceptions to address legal and ethical concerns, that should be clearly explained.

Response: All the data are now uploaded in a new file, but the details of the study subjects are hidden to maintain their privacy. All the gene data with NCBI links are already given in the previous submission.

• Authors should clearly describe participants’ inclusion and exclusion criteria, time of recruitment, how participants were selected (convenience sample?)

Response: This has been done in the current version of the manuscript. Please check page no. 4, line no. 92 to page no. 5, line no. 112.

• Authors should provide definitions/diagnostic criteria for described conditions/outcomes, including vitamin D deficiency.

Response: This has been done in the current version of the manuscript. Please check page no. 3, line no. 58 to 60 and page no. 6, line 118 to 122.

• The authors didn’t study other genetic determinants (other vitamin D associated genes), which could be possible confounders.

Response: This was the limitation of the study. As it was a part of an academic research project, we could not analyze other genetic determinants due to time and financial constraints. In addition, this has been explained in the current version of the manuscript. Please check page no.16, line no. 318 to 323 and page no.17, line no. 338 to 341.

• In statistical analysis, were data tested for normality (some data are reported as mean (SD) but appear not normally distributed). What about confounders?

Response: Yes, all data were tested for normality by the normal Q- Q plot of Shapiro- Wilk Normality Test (SPSS version 23 software). Mean (SD) of age, BMI, FBS, serum Ca++, creatinine, SGPT and albumin level of the study subjects and serum 25(OH)D level of the study population were apparently normally distributed. But serum 25(OH)D level of the controls were not normally distributed. The confounders could not be clearly understood for the mean serum 25(OH)D level of the controls as the number of the controls was only 10 and other genetic determinants could not be analyzed due to financial constraints.

• Study results and tables should be reported in the results section.

Response: This has been done in the current version of the manuscript. Please check page no. 9, line no. 195 and page no. 10, line no. 206.

• Tables require better organization with comparison of variables between the study and control groups.

Response: This has been done in the current version of the manuscript. Please check page no. 9, line no. 195, page no. 10, line no. 206 and page no. 13, line no. 245.

• In figure 1, it would be better to annotate the polymorphisms.

Response: This has been done in the current version of the manuscript.

• Figures 2 & 3 seem to be of little value as its data can be incorporated in tables or text.

Response: Figures 2 & 3 have been replaced by new figures and the previous data are incorporated in text in the current version.

• Discussion requires major revision, critically discussing main findings of study (without repeating results) and study implications.

Response: This has been done in the current version of the manuscript.

• In light of being a pilot, the current conclusion seems to overestimate study findings; authors should reduce the strength of their statement, confirming that this is a pilot study, and recommend further well-powered studies.

Response: Those have been changed in the current version. Please check page no. 16, line no. 326 to page no. 17, line no. 334.

• Please, revise the references to follow PLOS ONE style.

Response: This has been done in the current version.

• Authors should revise the manuscript for several errors in language, sentence structure, grammar, and punctuation.

Response: We have tried our best to follow proper language, sentence structure, grammar and punctuation in the current manuscript. 

Responses to Reviewer#1

• I have a major concern regarding the sample size of the study. With the current small sample size and the used study design, the study probably is underpowered. Was the sample size and power of the study was calculated prior starting the study?

Response: Thank you for your kind concern. This was a pilot study and none of the vitamin D influencing gene has ever been analyzed in Bangladesh. Moreover, as it was an academic research project, only 30 subjects as study population with low serum 25(OH)D level and 10 subjects sufficient serum 25(OH)D level as controls were included due to time and financial constraints. Therefore, we could not calculate the sample size and power of the study prior to the research. 

• In line 73 (in the introduction), Please remove DBP as GC is already mentioned and It is the gene encoding vitamin D binding protein (DBP). In addition, please include CYP27B1 gene with the other major common vitamin D genes as it is one of the significant genes involved in vitamin D metabolism (it is the gene encoding 1α-hydroxylase, the enzyme responsible for the 2nd step of activation of D in the kidney).

Response: This has been done in the current version of the manuscript. Please check page no. 4, line no. 74.

• Please mention the exclusion and inclusion criteria in the methods and the inter and intra-assay CV for the performed tests.

Response: This has been done in the current version of the manuscript. Please check page no. 5 line no. 98 to 112 and page no. 6, line no. 131 and 132. Except serum 25(OH)D level, the intra and inter assay CV of other parameters were not estimated. Other samples were analyzed for a single time as there was a scarcity of laboratory reagents due to financial constraints. This is because serum 25(OH)D level and genetic variants of CYP2R1 (rs10741657) were our study parameter and other parameters were estimated to confirm only the exclusion and inclusion criteria. 

• Please move Table 1 and 2 to the Results section.

Response: This has been done in the current version of the manuscript. Please check page no. 9, line no. 195 and page no. 10, line no. 206.

• Information in line 186-187 (in the results) is already mentioned in the methods, please remove.

Response: This has been removed in the current version of the manuscript. 

Responses to Reviewer#2

• First of all, I would like to congratulate the researchers for this very good work. The "CONCLUSION" section is also finished with a very good conclusion. However, the deficiency that I see in the study and other similar studies is that it is in the form of statistically researching the negativities associated with vitamin D deficiency and making a connection between them. This should provide an advantage if a selective mutation is dominant in geographic regions where this mutation is present. What advantage does vitamin d provide at lower levels? What protection does it provide for people in those areas? The answer to this question should also be explored in the comments. Natural selection never happens without a reason. When I look at it from the point of view of the country I live in, kidney stones come to mind. But I think this should be investigated. If researchers can add an answer and comment to this question in terms of their own country, I think that their work will be more beneficial than other studies. Beautiful original work worth publishing. The journal meets all publication criteria. I wish the researchers success in their new work.

Response: Thank you so much for your kind consideration. Actually, Bangladesh is a developing country and execution of such molecular research projects of any level is quite challenging. You may understand by the fact that this is the first genetic analysis which was carried out in Dhaka Medical College, Bangladesh. As, it was an academic study, it was to complete the whole research work in time considering the recent pandemic. Hence, your kind notes of appreciation have encouraged us to continue in this field of research in future. 

The aforementioned concerns have been included in the current version of the revised manuscript. Please check page no. 15 to 17.

Responses to Reviewer#3

• Grammar and sentence construction i nthe entire manuscript have to be revised/ improved

Response: Thank you for your kind concern. This has been revised in the current version. 

• What is the significance of yorus tidu if this has been reported in the literature already?

Response: This study has never been conducted in Bangladesh. As a pilot study, our goal was to determine the variants of CYP2R1 (rs10741657) gene and their association with low serum vitamin D level among Bangladeshi adults. The objectives have been elaborated in the current version, please check page no. 4, line no. 80 to 88. 

• inclusion and exclusion criteria not stated 

Response: This has been stated in the current version of the manuscript. Please check page no. 5, line no. 98 to page no.6, line no 122.

• how di you come up with the final nuber of subjects and how did you choose the initial of 54. please state clearly how you got yoru subjects

Response: Thank you so much again to give us a chance to clear about the fact. This was a pilot study and none of the vitamin D influencing gene has ever been analyzed in Bangladesh. Moreover, as it was an academic research project, only 30 subjects as study population with low serum 25(OH)D level and 10 subjects sufficient serum 25(OH)D level as controls were included due to time and financial constraints. This has been explained in the current version, please check page no. 6, line no. 122 to 136, and page no. 5, line no. 98 to 114.

• what was the significantce of all the other biochemical parameters that were taken apart form vitamin D levels

Response: This has been explained in the current version, please check page no. 5, line no. 107 to 111.

• for tables 1 and 2 make them reader friendly-remove too many lines and make it cleaner. yo ulaso did not specify the statistical significance betwenn the two groups of all the parameters

Response: This has been done in the current version of the manuscript. Please check page no. 9, line no. 195 and page no. 10, line no. 206.

• for table 3 you did not compare the results between groups. you only compared them within the groyp. it wuld be ebtter if you compared for exampe the genoype GG of affected and controls and the corresponding statistical significance

Response: Thank you for your concern, in table 3, only 2 controls had genotype GG, so the comparison between the groups was not done. In addition, the sample size was small and we took only 10 controls to see if there was any variation among the genotypes and alleles between study population and controls. For your kind concern we have added a new table to explain it. This has been explained in page no. 13, line no. 245 to 254 (Table 4). 

• i nthe discussion, do not repeat anymore your results but instead interpret and explain them. your tables hsould be self explanatory and in the discussion you will only give its importance

Response: Thank you so much for your kind advice. This has been corrected in the current version.

• what is the impact of the study?

Response: This has been explained in the current version. Please check page no. 16, line no. 313 to page no. 17, line no. 334.

• what is the implciation i nterms of treatment?

Response: This has been explained in the current version. Please check page no. 16, line no. 318 to page no. 17, line no. 334.

• make the discussion mroe concise

Response: This has been corrected in the current version accordingly. Please check page no. line no.

Thank you very much for considering our manuscript for revision.

Yours Sincerely,

Kazi Lutfar Rahman 

Email: klrahman@yahoo.com

---

## [Decision Letter · Decision Letter 1]

12 Oct 2021

PONE-D-21-25624R1Genetic Variations of CYP2R1 (rs10741657) in Bangladeshi adults with low serum 25(OH)D level- A Pilot StudyPLOS ONE

Dear Dr. Rahman,

Thank you for submitting your manuscript to PLOS ONE. The revised manuscript is much better. However, some minor, but important, comments have to be addressed.   You are kindly requested to respond to some concerns raised by Reviewer #3. Moreover, authors should briefly emphasize, in the results section, most important findings in the table (not just saying data are shown in tables). Last, the limitations should be moved to appear before the conclusion (could be integrated with the last paragraph in the discussion)  

We look forward to receiving your revised manuscript.

Kind regards,

Elsayed Abdelkreem, MD, PhD

Academic Editor

PLOS ONE

Journal Requirements:

Reviewers' comments:

Reviewer's Responses to Questions

**Comments to the Author**

1. If the authors have adequately addressed your comments raised in a previous round of review and you feel that this manuscript is now acceptable for publication, you may indicate that here to bypass the “Comments to the Author” section, enter your conflict of interest statement in the “Confidential to Editor” section, and submit your "Accept" recommendation.

Reviewer #2: All comments have been addressed

Reviewer #3: All comments have been addressed

2. Is the manuscript technically sound, and do the data support the conclusions?

Reviewer #2: Yes

Reviewer #3: Yes

3. Has the statistical analysis been performed appropriately and rigorously? 

Reviewer #2: Yes

Reviewer #3: Yes

4. Have the authors made all data underlying the findings in their manuscript fully available?

Reviewer #2: (No Response)

Reviewer #3: Yes

5. Is the manuscript presented in an intelligible fashion and written in standard English?

Reviewer #2: (No Response)

Reviewer #3: Yes

6. Review Comments to the Author

Reviewer #2: Since I think that this and similar studies, which have been done despite limited technical possibilities, should be supported, I would like to be approved for publication. Good luck to the researchers.

Reviewer #3: Thank you for your revisions. The manuscript has improved a lot i nterms of grammar, sentence construction, tables, and explanation of results

I still have some minor comments though. (I would like to apologize for all the typo errors during my initial comments).

1. You did not discuss the relevance of the significant differences in the biochemical parameters like sgpt, calcium, PT between subjects and controls. What were the reference values for these parameters? Or they were both in the normal ranges for both groups thats why they were not discussed anymore?

2. Clarification: were your subjects all healthy with incidental findings of low vitamin D levels? pls state that they were all healthy "meaning no other medical problem?"

3. I also wanted to know what is the implication in terms of management. Will you recommend vitamin D supplement to those who have low levels? If you will not give supplement, again what is the significance of the study if you will not be able to translate it to clinical practice? Likewise, did you consider checking if the subjects had concomittant disorders associated with low vitamin D levels? In your introduction you said that low vitamin D is related to several disorders. How can you correlate it with your results-- did you ask the medical histories of your subjects and check if these could be related to low vitamin D? (or are they all healthy?)

4. Will you also recommend to test for this polymorphism routinely among the population? If yes, what is the significance. Will this be harmful for their health?

5. For a better organization, I suggest created subheadings in your methodology. First discuss about the "study subjects" then the " biochemical analysis" and lastly "molecular analysis". Do not lump them altogether.

7. PLOS authors have the option to publish the peer review history of their article (what does this mean?). If published, this will include your full peer review and any attached files.

Reviewer #2: No

Reviewer #3: No

---

## [Author Response · Author response to Decision Letter 1]

24 Oct 2021

Responses to academic editor

• You are kindly requested to respond to some concerns raised by Reviewer #3. Moreover, authors should briefly emphasize, in the results section, most important findings in the table (not just saying data are shown in tables). Last, the limitations should be moved to appear before the conclusion (could be integrated with the last paragraph in the discussion)

Response: Thank you so much for your appreciation of the revised version. All the concerns raised by Reviewer #3 are responded accordingly in this current version of the manuscript. Most important findings of the table are emphasized in the current version, please check page no. 09, line no. 194 to 195 and page no. 11, line no. 232 to page no. 12, line no. 245. The limitations section is moved to appear before the conclusion as you had suggested, please check page no. 17, line no. 344 to 349.

According to the journal requirement, all the references have been revised carefully. In the first round of revision of the manuscript, irrelevant references were removed and only relevant references were kept and added. In this round of revision, we have carefully checked that no changes and retractions have been made to the references. Currently, the references which have been used all are available and not retracted.

In the first revised version, we removed and added some of our references which were not mistakenly elaborated in previous rebuttal letter and now those are added here,

Reference:

Zgheib, Nathalie & Arabi, Asma & Mahfouz, Rami & El-Hajj Fuleihan, Ghada. (2013). OC002—Cyp2r1 Genetic Polymorphisms Are Associated With Lower 25-Hydroxy Vitamin D Levels In Lebanese Subjects. Clinical Therapeutics. 35. e1. 10.1016/j.clinthera.2013.07.003. 

Status: Removed 

Comment: The information for which it was that had been modified and the modified information is available in no.15 article of the current version. 

Reference:

Holick MF, Binkley NC, Bischoff-Ferrari HA, Gordon CM, Hanley DA, Heaney RP, et al. Evaluation, treatment, and prevention of vitamin D deficiency: An endocrine society clinical practice guideline. J Clin Endocrinol Metab. 2011;96: 1911–1930. doi:10.1210/jc.2011-0385

Status: added 

Comment: It was used to define the refence value of serum 25(OH)D level, (ref no.4 in current version)

Reference:

Binkley N, Ramamurthy R, Krueger D. Low vitamin D status: Definition, prevalence, consequences, and correction. Endocrinol Metab Clin North Am. 2010;39: 287–301. doi:10.1016/j.ecl.2010.02.008

Status: added 

Commenet: It was used to define the low serum 25(OH)D level, (ref no.5 in current version)

Reference:

Ahmed B, Shiraji KH, Chowdhury MHK, Uddin MG, Islam SN, Hossain S. Socio-economic Status of the Patients with Acute Coronary Syndrome: Data from a District-level General Hospital of Bangladesh. Cardiovasc J. 2017;10: 17–20. doi:10.3329/cardio.v10i1.34357

Status: added 

Comment: It was used to determine the socioeconomic status of the study subjects, (ref no.16 in current version)

Reference:

Islam MA, Alam MS, Sharker KK, Nandi SK. Estimation of Solar Radiation on Horizontal and Tilted Surface over Bangladesh. Comput Water, Energy, Environ Eng. 2016;05: 54–69. doi:10.4236/cweee.2016.52006 

Status: added 

Comment:

It was used to determine sun exposure according to geographic location of Bangladesh (ref no.27 in current version)

Finally, the reference numbers were rearranged according to the article rearrangement during revision.

Responses to Reviewer#3

• You did not discuss the relevance of the significant differences in the biochemical parameters like sgpt, calcium, PT between subjects and controls. What were the reference values for these parameters? Or they were both in the normal ranges for both groups that’s why they were not discussed anymore?

Response: Thank you so much for your concern to make a better version of this article. All the biochemical parameters were within the normal ranges among both the study population and the controls, so they were not discussed anymore. Here, we have added the chart that we followed to ensure the normal range of all the biochemical parameters.

Reference values of different parameters

 Parameters Normal range

1. Serum calcium 8.50-10.30 mg/dl (Automate 

 Biochemistry Analyzer)

2. Serum albumin 3.40 – 5.00 gm/dl (Automated Biochemistry Analyzer)

3. Fasting blood glucose 3.30 – 6.11 mmol/L (Automated Biochemistry 

 Analyzer)

4. Serum Creatinine Male: 0.60 1.50

 Female: 0.40 1.20

 Child: 0.20 0.70 (Automated Biochemistry Analyzer)

5. Serum SGPT < 40 U/L (Automated Biochemistry Analyzer)

6. Serum 25(OH)D Deficiency:<20ng/ml 

 Insufficiency: 20-29.9ng/ml

 Sufficiency: ≥30ng/ml

7. Prothrombin time 11-16 second.

• Clarification: were your subjects all healthy with incidental finding of low vitamin D levels? pls state that they were all healthy “meaning no medical problem?” 

Response: Thank you for your concern. Yes, all the subjects were healthy and based on your suggestion we have added this clarification in the methodology. Please check page no.5, line no.111.

• I also wanted to know what is the implication in terms of management. Will you recommend vitamin D supplement to those who have low levels? If you will not give supplement, again what is the significance of the study if you will not be able to translate it to clinical practice? Likewise, did you consider checking if the subjects had concomitant disorders associated with low serum vitamin D level? In your introduction you said that low vitamin D is related to several disorders. How can you correlate it with your result- did you ask the medical histories of your subjects and check if these could be related to low vitamin D? (or are they all healthy?)

Response: Thank you for your concern. We began the process by taking the medical history of the study subjects. Their medical history did not reveal any pathological conditions related to low serum 25(OH)D level. In addition to this, we investigated serum calcium level to determine any abnormalities associated with calcium homeostasis. We had a financial constraint for which we were not able to assess serum PTH and other investigations which might be associated with any vitamin D related abnormalities. Since it was a pilot study, the sample size was small. For better results, we need to perform this study on a larger scale. In this pilot study, we have observed that people with low serum 25(OH)D level can perform their physiological functions without any hardship and it is evidenced by the investigation results and also by the history of having no vitamin D related disorders. As all the study subjects were apparently healthy individuals, so it can be hypothesized that due to natural selection, which could be based on the geographical location and sun exposure, variants of rs10741657 (GG, GA) can perform their physiological functions with low serum 25(OH)D level without any need for any kind of vitamin D supplements. But again, to come to a better conclusion, we need further studies with larger sample size and other vitamin D related biochemical analysis. The major implication of this study is to consider the genetic influence in order to redefine the cutoff value of low serum vitamin D level. All these are explained elaborately in the discussion and the conclusion portion of the manuscript, please check page no.16, 17 and 18.

• Will you also recommend to test for this polymorphism routinely among the population? If yes, what is the significance. Will this be harmful for their health?

Response: Since it was a pilot study with a small sample size, we need to further assess to confirm the genetic influence of this polymorphism on vitamin D level on a larger scale. Only then we can recommend if this polymorphism can routinely be done among the population or not. If the precautions are properly taken, it will not be harmful for the health. It has been added in the current version please check page no.17, line no.339 to 342.

• For a better organization, I suggest created subheadings in your methodology. First discuss about the ‘Study Subjects’ the “biochemical analysis” and lastly “molecular analysis”. Do not lump them altogether.

Response: Thank you for your concern, these have been done in the current version. Please check page no. 5 line no. 116, page no. 6 line no. 132 and page no. 7 line no. 141.

---

## [Decision Letter · Decision Letter 2]

8 Nov 2021

Genetic Variations of CYP2R1 (rs10741657) in Bangladeshi adults with low serum 25(OH)D level- A Pilot Study

PONE-D-21-25624R2

Dear Dr. Rahman,

We’re pleased to inform you that your manuscript has been judged scientifically suitable for publication and will be formally accepted for publication once it meets all outstanding technical requirements.

Kind regards,

Elsayed Abdelkreem, MD, PhD

Academic Editor

PLOS ONE

Additional Editor Comments (optional):

Reviewers' comments:

Reviewer's Responses to Questions

**Comments to the Author**

1. If the authors have adequately addressed your comments raised in a previous round of review and you feel that this manuscript is now acceptable for publication, you may indicate that here to bypass the “Comments to the Author” section, enter your conflict of interest statement in the “Confidential to Editor” section, and submit your "Accept" recommendation.

Reviewer #3: All comments have been addressed

2. Is the manuscript technically sound, and do the data support the conclusions?

Reviewer #3: Yes

3. Has the statistical analysis been performed appropriately and rigorously? 

Reviewer #3: I Don't Know

4. Have the authors made all data underlying the findings in their manuscript fully available?

Reviewer #3: Yes

5. Is the manuscript presented in an intelligible fashion and written in standard English?

Reviewer #3: No

6. Review Comments to the Author

Reviewer #3: (No Response)

7. PLOS authors have the option to publish the peer review history of their article (what does this mean?). If published, this will include your full peer review and any attached files.

Reviewer #3: No

---

## [Editor Report · Acceptance letter]

10 Nov 2021

PONE-D-21-25624R2 

Genetic Variations of *CYP2R1* (rs10741657) in Bangladeshi adults with low serum 25(OH)D level -A Pilot Study 

Dear Dr. Rahman:

I'm pleased to inform you that your manuscript has been deemed suitable for publication in PLOS ONE. Congratulations! Your manuscript is now with our production department. 

Kind regards, 

on behalf of

Dr. Elsayed Abdelkreem 

Academic Editor

PLOS ONE